# User Experience during an Immersive Virtual Reality-Based Cognitive Task: A Comparison between Estonian and Italian Older Adults with MCI

**DOI:** 10.3390/s22218249

**Published:** 2022-10-27

**Authors:** Marta Mondellini, Sara Arlati, Helena Gapeyeva, Kairi Lees, Ingrid Märitz, Simone Luca Pizzagalli, Tauno Otto, Marco Sacco, Anneli Teder-Braschinsky

**Affiliations:** 1Institute of Intelligent Industrial Technologies and Systems for Advanced Manufacturing (STIIMA), National Research Council (CNR), Via Previati 1/E, 23900 Lecco, Italy; 2Clinic of Medical Rehabilitation, East Tallinn Central Hospital, Ravi Street 18, 10138 Tallinn, Estonia; 3Department of Mechanical and Industrial Engineering, Tallinn University of Technology (TalTech), Ehitajate tee 5, 19086 Tallinn, Estonia

**Keywords:** cognitive training, assessment, technology acceptance, mild cognitive impairment, elderly

## Abstract

Mild cognitive impairment (MCI) is an early stage of cognitive abilities loss and puts older adults at higher risk of developing dementia. Virtual reality (VR) could represent a tool for the early assessment of this pathological condition and for administering cognitive training. This work presents a study evaluating the acceptance and the user experience of an immersive VR application representing a supermarket. As the same application had already been assessed in Italy, we aimed to perform the same study in Estonia in order to compare the outcomes in the two populations. Fifteen older adults with MCI were enrolled in one Rehabilitation Center of Estonia and tried the supermarket once. Afterwards, they were administered questionnaires aimed at evaluating their technology acceptance, sense of presence, and cybersickness. Estonian participants reported low side effects and discrete enjoyment, and a sense of presence. Nonetheless, their intention to use the technology decreased after the experience. The comparison between Italian and Estonian older adults showed that cybersickness was comparable, but technology acceptance and sense of presence were significantly lower in the Estonian group. Thus, we argue that: (i) cultural and social backgrounds influence technology acceptance; (ii) technology acceptance was rather mediated by the absence of positive feelings rather than cybersickness.

## 1. Introduction

Dementia is a pathological condition, generally chronic or progressive, caused by a disease of the brain. The causes of dementia are diverse: the most common is Alzheimer’s disease [1], i.e., a pathological condition characterized by the deposition of β-amyloid plaques and neurofibrillary tangles, which collect at the extracellular and intracellular levels, disrupting nervous cells’ functions [2]. The development of dementia in aging people proceeds insidiously and, over the years, leads from a state of cognitive normality to progressively severe stages of intellectual and motor dysfunctions [3]. Nowadays, it is estimated that more than 50 million people worldwide have dementia, and this number is expected to triple by 2050 [4].

For more than 40 years, clinicians have been working on the problem of defining the boundaries between normal age-related cognitive impairment and the intermediate levels of deterioration that precede the onset of dementia [3]. The term “mild cognitive impairment” (MCI) arises precisely in this research context to indicate individuals who do not have dementia but problems with memory, language, thinking, or judgment that are greater than what we would expect with normal aging [5]. The term has been used more and more frequently over the years in the field of clinical research, reflecting the shift of interest from research and treatment of dementia to the early diagnosis of its first symptoms [6]. According to associations for Alzheimer’s patients, such as the Alzheimer’s Society of the UK [7], people with MCI are at a higher risk of developing dementia (10–15% of people with MCI); furthermore, it is estimated that between 5 and 20% of people aged over 65 have MCI. People with intellectual disabilities are one of the most vulnerable populations: they often do not have a job, do not live in their own home, and their life is managed by third parties [8]; moreover, the economic and psychological cost for relatives and caregivers is high.

Because of these reasons, the early assessment—although complicated [9]—and treatment of MCI could play a key role in preventing further cognitive decline and all its associated issues.

Virtual reality (VR) may represent an effective and adaptable tool able to offer the assessment and the training of cognitive abilities in older adults [10]. VR is defined as an artificial, virtual, and viewer-centered experience that blocks out the physical world, replacing it with a computer-generated one [11,12]. It provides the opportunity to practice, in a safe condition, daily-life activities that cannot normally be experienced within the clinical setting, e.g., training attention while crossing a street [13]. This offers an ecological context to the patient and allows a better measure of everyday function [14]. In addition, VR environments can be created and tailored on the user: in the real world, a task or request could be too difficult to accomplish, and it could overwhelm the user, frightening them or making them feel humiliated. Virtual worlds can propose exercises of increasing difficulty, so as to offer a balanced challenge and avoid frustration [15]. Another important aspect is that through VR it is possible to propose a series of similar exercises that reproduce activities of daily life, facilitating these users, even with intellectual disabilities, to generalize the acquisition of skills in the real world [16]. Keeping in mind that users with MCI are mostly older adults and possibly suffering from physical difficulties, VR-based assessment or training can be offered through accessible interfaces [17]. Finally, gamification techniques, namely the use of game design elements in non-game contexts, immediate feedback on users’ performance, and a high degree of realism and subsequent immersion enhance motivation and encourage higher numbers of repetitions of the same exercise, also fostering treatment compliance [18].

In spite of these advantages, it is also true that VR may also have some drawbacks. The most known is cybersickness, i.e., a series of symptoms arising during the experience in VR, which may entail: nausea, vision discomfort, headache, dizziness, and disorientation [19]. It emerges as a consequence of sensory feedback mismatch, vergence and accommodation conflict, vection, and (lack of) navigation control [20]. Cybersickness and other potential VR-related discomforts, like the weight of the headset and the inability to control or comprehend what is happening in the virtual world, may negatively affect the acceptance of VR technology and lead to a refusal to use it [21].

Given these points, assessing the acceptance of a newly developed tool represents an essential step for the validation of any VR-based solution.

In the present work, the user experience of an immersive VR application dedicated to the training and assessment of cognitive abilities in patients with MCI of Rehabilitation Center of Estonia was evaluated. Fifteen Estonian older adults with MCI carried out a shopping task in a virtual supermarket, supported by an occupational therapist who assisted and supervised the VR experience. Technology acceptance, and the variables influencing it, i.e., sense of presence and cybersickness [22] were evaluated. Objective performance was also recorded.

The same virtual environment and tasks have been validated in terms of user experience in older adults with objective and subjective cognitive decline in Italy (Section 2). However, as different backgrounds at cultural, social, and economic levels [23,24,25] involve a different attitude toward technology [26,27], reproducing the study was believed essential to ensure a positive experience also in the Estonian end users.

The remainder of this paper is organized as follows: Section 2 presents related work and briefly describes the outcomes of the study carried out with older adults from Italy; Section 4 describes the population, the equipment, and the methods used to carry out the current study; Section 5 reports the study outcomes and the comparison between the results obtained in Italy and Estonia; finally, Section 8 draws the conclusions and outlines future works.

## 2. Previous Work

Recently, the use of eXtended Reality (XR) technology has emerged as a promising solution for the management and rehabilitation of cognitive and motor functions [28]. Table 1 reports the definitions of main XR technologies and some examples of their application in the medical context, with regard to dementia management in particular.

Among these technologies, VR is the most common [28], and its use to assess cognitive abilities and administer cognitive stimulation is raising researchers’ attention [37]. A recent review [30] on these topics showed that older adults found the proposed tasks challenging and enjoyable. Additionally, in most of the cases, no adverse events have been reported. Moreover, VR-based applications demonstrated potential in discriminating cognitive deficits from normal cognition and in promoting motivation throughout cognitive training sessions. Nonetheless, evidence is still preliminary (i.e., only 9 studies were included), and the same authors concluded that designing a VR environment that excludes unwanted symptoms and harm to vulnerable individuals like patients with MCI and dementia remains essential. In this context, our research group developed a Virtual Supermarket to evaluate and train visuospatial abilities in patients with MCI. The choice of the supermarket was driven by the need to integrate some cognitive tasks into an activity of daily living that was familiar for most older adults [38].

Some other works foresaw the use of a supermarket environment for cognitive purposes. Ouellet et al., developed the Virtual Shop, an immersive VR environment to assess everyday memory [14] and tested its validity in a group of young (age: 21.65±2.46) and older adults (age: 68±5.03). This approach was proven feasible and able to discriminate between the two age groups. Moreover, the performances in the virtual task were correlated with the outcomes of standard cognitive tests. The same discrimination ability was also found in a non-immersive supermarket application deployed for Android tablets [39]. VR has also been employed to train cognitive abilities with positive outcomes on global cognition, even when compared to active control conditions [37]. Additionally, in this case, supermarket scenarios have been successfully employed in both 2D and 3D immersive environments [29,40].

The application presented in this work (see Section 4.2.1) has foreseen a first validation phase enrolling healthy young adults (age: 28.75±3.65) to exclude the arousal of cyber-sickness, assess its usability, and improve the interaction and intuitiveness of the interfaces [41]. Afterwards, it was administered once to a group of Italian older adults with MCI or subjective cognitive decline (SCD) (age: 71.4±5.85; gender: 38F/19 M) [42] to assess its acceptance and usability in a group of target users. Such a study revealed that the Virtual Supermarket was well-accepted, enjoyable, and engaging for older adults, with only minor differences between the two groups of participants. No severe cyber-sickness arose, with the exception of one participant (out of 58) who wanted to interrupt the experience.

Given this positive experience, the Virtual Supermarket was slightly modified and used in Estonia at East Tallinn Central Hospital as a potential tool for the diagnosis of MCI and cognitive training. This version has been termed VSEE-modified. A few grocery items were modified to fit the Estonian supermarket packages, and grocery items’ labels with prices were introduced. Such modifications were minimal. However, given Estonian older adults’ different cultural, social, and economic backgrounds with respect to Italian individuals, a new study aimed at assessing the acceptance and usability of immersive VR was believed to be needed to ensure acceptance of the Virtual Supermarket in Estonian end users [43].

## 3. Aims

This work has two main aims:1.To assess the user experience of an immersive VR-based application in a group of Estonian older adults with MCI, in order to evaluate whether it could be used, in the clinical practice, as a tool for the assessment and the training of visuospatial abilities;2.To compare the Estonian older adults’ user experience, both at subjective (i.e., via standard questionnaires) and objective (i.e., performance) levels, to Italian older adults’ UX, obtained in a previous study [42]. Such a comparison could be meaningful to verify whether a different social, cultural, and economic background could (i) impact UX in general and (ii) assess which factors are mainly influenced.

To the authors’ knowledge, this is the first study comparing user experience while using VR in two different countries. We hypothesized that user experience would have been good among Estonian users, and mostly comparable to the outcomes obtained in Italy. In addition, given Estonians’ higher familiarity with the technology, we also expected to detect higher perceived ease-of-use and lower anxiety during the experience.

## 4. Materials and Methods

### 4.1. Participants

Participants were recruited among the attendees of East-Tallinn Central Hospital. Fifteen older adults were enrolled in the present study. Inclusion criteria were: age ≥65 and diagnosis of MCI assessed via Montreal Cognitive Assessment (MoCA) test (Section 4.4.1). Exclusion criteria were: dementia or psychiatric disease; the presence of significant functional or visual impairment that could prevent the use of the VR usage; cardiovascular disease or seizure; history of motion sickness, or any other health condition that could prevent the experience in VR. Stroke did not represent an exclusion criterion if not with consequences leading to the functional problems reported before.

Written informed consent was obtained before participation in the study. The study was carried out in agreement with the Helsinki declaration and approved by the Estonian National Institute for Health Development Ethical Committee (approval no. 167 of 12 December 2019).

### 4.2. Equipment

#### 4.2.1. The Virtual Supermarket

The Virtual Supermarket in its VSEE-modified version was developed in Unity and deployed for the HTC Vive headset. It is composed of two main scenes covering a game area of about 4 m × 3 m: the aisle scene, containing supermarket shelf units filled with grocery items, and the cash-register scene, in which the users can pay for the items they had picked. In the aisle scene, patients were asked to do the shopping, picking all the items presented on a list and putting them in a cart. The cart was fixed along one of the two short sides of the room, and the list was anchored on it. To pick an item, patients had to use the HTC Vive controller; in particular, they had to collide with the controller with the product to pick and then press the back trigger button to grab it and drag it around. The release of the trigger also caused the release of the item. The collision of the controller with any of the items on the supermarket shelf was signaled via the vibration of the controller. As mentioned, these interactions were defined in preliminary studies enrolling healthy young adults. Each item on the shelf was labeled with its name and price (Figure 1). As an additional hint for the patients, two aisle signs containing the names of the items on the list were displayed above the cart.

The positions of the grocery items on the shelves were not varied throughout the training, whereas the list of the items to pick changed from time to time. All the items could be grabbed and dropped with no restrictions; whenever an item placed in the cart corresponded to one on the list, such an item’s name was marked in green on the list. In the cash-register scene (Figure 2), the patients were required to pay a randomly generated amount by selecting banknotes and coins to avoid any change.

#### 4.2.2. Hardware

As mentioned, the HMD used for this study was from HTC Vive. The HTC Vive VR kit comprises an HMD, two controllers, and two infrared emitters that track the motion of the HMD and the controllers in the game area. The controllers are symmetrical and could be used with both the right and the left hands. The Virtual Supermarket ran on a Windows PC with Intel(R) Core(TM) i5-6500 CPU (Intel Corporation, Santa Clara, CA, USA).

### 4.3. Study Protocol

All participants followed the same protocol. First, participants completed an anamnestic questionnaire, and their cognitive level was measured with the Montreal Cognitive Assessment. Moreover, their cybersickness symptoms and their intention to use the virtual reality system were measured (Section 4.4). After completing this initial phase, the researcher helped the patient wear the headset and headphones and provided information about the tasks. Each participant was asked to complete the shopping once, picking the 8 items on the list in the aisle scene and then performing the payment. The whole experience time was not fixed, but variable according to the subjective performances and skills. At the end of the test, participants were required to complete questionnaires (Section 4.4) to evaluate their experience during the immersion (see Section 4.4). Spontaneous user comments were transcribed by the researcher to also collect qualitative data about the user experience.

### 4.4. Measures

The outcomes were collected at two time points: before the experience in VR (Section 4.4.1) and immediately after (Section 4.4.2). In addition, participants’ spontaneous comments were recorded throughout the experience by the occupational therapist supervising the trial, who transcribed all relevant remarks.

#### 4.4.1. Baseline Assessment

Before the VR experience, the enrolled older adults completed an anamnestic questionnaire and a cognitive screening test; a standard questionnaire to assess the presence of cybersickness symptoms; and some questions to evaluate their intention to use VR technology in the future.

The anamnestic questionnaire collected the following data: sex, age, education, occupation, nationality, level of computer experience, and level of knowledge of VR. To collect the other data, the following instruments were used.

The MoCa is a tool administered by health professionals for the detection of mild cognitive impairment and was used in this study to confirm MCI [44]. It requires 10 min and assesses several cognitive domains (e.g., visuospatial/executive functions, memory, attention, language, and orientation). The maximum score is 30, and a score of 26 or above is considered normal.

The Simulator Sickness Questionnaire (SSQ) [45] is a tool for evaluating the potential side-effects that occur during the experience in virtual environments. SSQ consists of 3 subscales: nausea (SSQ-N), oculomotor disorders (SSQ-O), and disorientation (SSQ-D); each subscale includes 7 symptoms that can be classified on a Likert scale from 0 (absent) to 3 (severe). This questionnaire was originally created to evaluate the adverse effects resulting from the use of driving simulators; even though it has not a very high validity [46,47], this tool is still widely used for the evaluation of cybersickness in virtual environments and in contexts different from driving. Therefore, it was used in the present work to offer a comparison with the majority of other studies with VR. The ranges of scores on nausea, oculomotor disturbances, disorientation, and Total Score are 0–200.34, 0–159.18, 0–292.32, and 0–235.62, respectively. The authors report that SSQ is a particularly useful tool for understanding the general severity of side effects [45].

Lastly, before the VR experience, older adults answered three questions about their intended use of the system in the future. These questions were taken from the subscale “behavioral intention (BI)” of the questionnaire TAM3 [48], in which answers are coded on a 7-item Likert scale, ranging from “strongly disagree” (1) to “strongly agree” (7). Since the final score is the mean of the three items, the range of scores for this scale is 1–7.

#### 4.4.2. User Experience Evaluation

The experience in the Virtual Supermarket was evaluated using both quantitative and qualitative data. To assess the experience quantitatively, in terms of cybersickness, sense of presence, and technology acceptance, the following instruments were used: SSQ, the International Test Commission—Sense of Presence Inventory (ITC-SOPI) [49], and Technology Acceptance Model 3 questionnaire [48].

ITC-SOPI is a questionnaire that investigates the sense of presence experienced during the virtual experience, namely the sense of “being there” in the computer-generated world. This tool is composed of 4 subscales that investigate the spatial presence, i.e., the sense of being immersed in the mediated environment and of control over it (SP, 23 items), engagement or the sense of being psychologically involved in the experience (ENG, 18 items), naturalness, namely how much the mediated environment seems realistic (NAT, 8 items) and side-effects (SE, 6 items). The total score of each is the mean of its items, so the range score is 1–5 [49]. The analyses described in [49] demonstrated that the ITC-SOPI is a reliable and valid tool. TAM3 is a questionnaire based on an information systems theory that models how individuals perceive a technology, and, eventually, accept and use it in daily life [48]. For this study, only the following TAM3 subscales were used:Perceived ease of use (PEOU), i.e., the degree to which a person believes that the use of a system is easy (4 items),Computer anxiety (CANX), i.e., the individual apprehension that is perceived towards technology (4 items),Perceived enjoyment (ENJ), i.e., the extent to which the activity of using a system is perceived as enjoyable, regardless of the performance of the system itself (3 items),Behavioral intention (BI) (3 items), as before the experience.

The range of scores for each scale is 1–7. In [48], all constructs at each time period have exhibited strong psychometric properties and have satisfied the criteria of reliability and convergent and discriminant validity.

#### 4.4.3. Performance Variables

To assess each participant performance, the following objective variables were recorded: the time—in minutes—needed to complete the supermarket and payment exercise, the number of times an item has been dropped, dropped and collected, and the number of errors made during payment.

### 4.5. Statistical Analysis

All statistical analyses were performed using IBM SPSS v.26 software (IBM Corp, Armonk, NY, USA) [50]. Asymmetry and kurtosis were observed for all the evaluated variables to verify any variation from their normal distribution. We considered non-normal values of asymmetry higher than |2| and kurtosis higher than |3| [51]. The internal consistency of TAM3 subscales has been verified by using Cronbach’s alpha, as some items were left out (according to what was reported in [48]).

Participants’ characteristics and performance were analyzed using descriptive and frequency analyses. Due to the small sample size, the Wilcoxon signed-rank non-parametric test was run to compare the intention to use the system before and after the immersive experience [52].

Correlations were measured between sense of presence, acceptance of the system, cybersickness, and performance (seconds needed to complete tasks and number of errors) using the non-parametric Spearman index. To measure the strength of a relationship between two variables excluding the effect of one or more other variables, non-parametric partial correlations were performed.

Cybersickness, acceptance, and sense of presence were compared between the Estonian and the Italian participants running the Mann–Whitney U test for independent samples.The same test was run to compare Italian and Estonian users’ performance (i.e., the time taken to complete the first shopping task and the errors committed in the first trial by Italian participants vs. the time and the errors of Estonian participants in their only trial). The level of significance was set at *p* < 0.05.

## 5. Results

All participants in this study were retired from work at the time of the testing. The enrolled group consisted of one man and 14 women, of which 13 were Estonians and 2 were Russians living in Estonia, with an average age of 75.73±6.36. Seven participants reported having no computer experience, seven having some basic knowledge, and one being moderately experienced. Fourteen declared to be completely inexperienced in VR, and one to have basic knowledge in VR. MoCA questionnaire scores indicated that patients reported a minimum score of 21 and a maximum of 25, with an average score of 22.93±1.44. This score showed that the cognitive level of the Estonian participants is comparable to that of the Italians’ participants (Italian patients’ Mini-Mental State Examination: 26.63) [53,54]. The subscales of the TAM3 questionnaire was highly reliable, with α=0.94 for behavioral intention, α=0.835 for post-test behavioral intention, α=0.81 for perceived ease-of-use, α=0.954 for anxiety and α=0.974 for enjoyment.

### 5.1. Subjective and Objective Variables’ Characteristics

As mentioned before, all variables were normally distributed, with the exception of some subscales of SSQ: pre-test nausea (kurtosis = 3.11), post-test oculomotor symptoms (kurtosis = 4.231), post-test disorientation (kurtosis = 3.01), and post-test nausea (asymmetry = 2.02). All the other descriptive and frequency analyses are reported in Figure 3 and Table 2.

### 5.2. Pre- and Post-Experience Differences

The Wilcoxon signed-rank test revealed that intention to use the system was higher before (Md=4, N=15) than after (Md=2.67, N=15) the experience, with z=−2.79 and p=0.005. Regarding SSQ scores, no differences in the total score and in the subscales’ score were found between pre- and post-experience.

### 5.3. Correlation between Subjective Variables

With respect to cybersickness, all simple correlations of each SSQ subscale and ITC-SOPI subscale of side effects are reported in Appendix A.

Using the partial correlation analysis on the collected data, the following correlations are observed: pre-test nausea and oculomotor disturbances ($\rho_{s}\left(6\right)=0.72,p<0.05$); pre-test nausea and post-test disorientation ($\rho_{s}\left(6\right)=-0.75,p<0.05$), pre-test nausea and post-test oculomotor disturbances ($\rho_{s}\left(6\right)=0.84,p<0.01$), pre-test nausea and total symptoms ($\rho_{s}\left(6\right)=0.87,p<0.005$), pre-test oculomotor disturbances and post-test disorientation ($\rho_{s}\left(6\right)=0.74,p<0.05$), post-test disorientation and total symptoms ($\rho_{s}\left(6\right)=0.81,p<0,05$), pre- and post-test disorientation ($\rho_{s}\left(6\right)=0.9,p<0.005$), and pre-test total score and post-test disorientation ($\rho_{s}\left(6\right)=-0.81,p<0.05$).

Regarding sense of presence, ITC-SOPI subscales correlated, with the exception of the subscale side effects, but only spatial presence and naturalness showed a strong correlation when excluding the influences of engagement ($\rho_{s}\left(11\right)=0.80,p=0.001$). Simple correlations of these variables are reported in Appendix A.

Some aspects of cybersickness before and after the immersion in the VR environment correlated with TAM3 subscales, but only with simple correlations (Appendix A).

Simple correlations between TAM3 subscales emerged, as well as between TAM3 subscales and ITC-SOPI variables, but running partial correlations, no variables resulted as related. Simple correlations are reported in Appendix A.

Cognitive status (MoCA scores) was related to post-experience nausea ($r_{s}=0.62$, p<0.05) and engagement ($r_{s}=0.53,p<0.05$), but only with simple correlation.

### 5.4. Correlation between Subjective Variables and Performance

With respect to objective data, pre-immersion oculomotor disturbance was inversely related to the number of items dropped and collected ($\rho_{s}=-0.58,p<0.05$), post-test nausea was related to wrong objects selected ($\rho_{s}=0.57,p<0.05$), and seconds needed to complete aisle scene related to the seconds needed to complete the cash register scene ($\rho_{s}=0.73,p<0.005$).

By keeping the influences of the other cybersickness variables on the first two correlations under control, pre-experience oculomotor disturbances and drop errors, and post-experience nausea and errors were not correlated. Furthermore, the participants’ performances were not correlated with their characteristics: either age, MoCA score, or technological expertise.

### 5.5. Spontaneous Comments and Clinicians’ Observations

The qualitative data collected consisted of participants’ spontaneous comments and clinicians’ observations during the interaction. Three participants reported that they appreciated the system and the exercise. Specifically, two patients said they perceived the experience as interesting and educational, while another said the experience was enjoyable and s/he would recommend it to others. Conversely, one patient stated “such a game is good for children, not for older adults”, while another reported that the controller was uncomfortable and that s/he could not find one of the items listed on the shelf. In general, occupational therapists observed that the patients needed a lot of support or guidance. In particular, three patients had difficulty using the controller, both in the shopping scene and in the payment scene. For two participants, it was particularly difficult to identify the objects on the list, while one patient found it hard to select the objects on higher shelves.

### 5.6. Differences with the Italian Participants

No difference in cybersickness between the Italian and Estonian participants was present (Figure 4). Instead, differences emerged in the TAM3 and ITC-SOPI subscales (Figure 4), with z=−3.20,p=0.001 for perceived ease-of-use, z=−2.64,p<0.01 for anxiety, z=−2.07,p<0.05 for enjoyment, z=−3.73,p=0.000 for behavioral intention, z=−3.06,p<0.005 for spatial presence, z=−2.47,p<0.05 for engagement, and z=−2.79,p=0.005 for naturalness.

Regarding objective performance, no difference in errors emerged, but significant differences occurred in the time needed to complete the aisle scene (z=−3.64,p=0.00). Indeed, Italian participants took an average time of 4.38 min to complete the aisle scene (SD: 2.69; min/max time 1.18/16.75).

## 6. Discussion

This study investigated the user experience of Estonian older adults with MCI while using an immersive VR application dedicated to cognitive training and assessment. Our findings supported that the Virtual Supermarket experience was generally positive. However, contrary to what we expected, we found that sense of presence and technology acceptance were significantly lower in Estonian older adults compared to their Italian peers.

The participants of the Rehabilitation Center of Estonia consisted mostly of women with MCI (range for MCI 19–25). Cognitive impairment was associated with longer time needed to complete the shopping scene, but not with the time needed to pay or the number of errors. As expected, the times needed to complete the aisle scene and the payment correlated positively. The data indicated a relationship between the number and type of errors committed and cybersickness. Still, this association is usually strongly influenced by other factors, such as the perception of exercise intensity and the motivation [55]. A more in depth investigation with a higher number of participants, along with an increased number of variable considered could lead to a better understanding of the performance data.

Similarly, the subscales of the ITC-SOPI questionnaire also positively correlate with each-others, with the exception of the side effects subscale. These results partially confirm what Lessiter [49] reported, and, again, it is possible that the lack of correlation between spatial presence and side effects is due to the sample size.

Although correlations do not persist running partial correlations, some associations that should probably be observed in a larger sample emerged in our work. First of all, symptoms of disorientation (pre- and post-test) were associated with less enjoyment, as also reported in [56]. It should also be noted that this is the SSQ subscale most associated with cybersickness in VR (unlike what is reported in the use of simulators [46], where nausea is generally predominant). In future studies, we suggest adding a self report tool, such as the VRSQ, to measure the physical side effects of immersion in VR.

Furthermore, post-test cybersickness is inversely related to perceived ease-of-use as well as to the behavioral intention, in accordance with the results of [22]. Pre-test behavioral intention is also correlated with post-test behavioral intention, and the latter is correlated to a positive subjective experience (enjoyment, engagement, spatial presence), confirming that the intention to use the proposed technology is strongly influenced by intrinsic and subjective factors [57]. As also reported in [58], fun was associated with ease of use. Finally, cognitive level and engagement are positively related; this could be due to the influence on both variables of a common factor, e.g., consciousness [59].

Performance data related to cybersickness in an unclear way and only with simple correlations. Thus, it is necessary to evaluate this association in the future. Nonetheless, this data is in line with the conclusions of other studies in which a general effect of VR exposure (slightly mediated by cybersickness) on performance is hypothesized [60]. Another aspect that could affect the results may be associated with language function and praxis [61], namely neurological damages caused by stroke, which was not an exclusion criterion for this study.

Regarding differences with the Italian group, the Estonian participants scored significantly lower in all TAM3 and ITC-SOPI subscales, with the exception of the anxiety subscale (CANX) in which Estonian patients reported higher scores. Instead, in all cybersickness-related subscales (SSQ and ITC-SOPI side effects), no differences emerged.

It is interesting to note that, although cybersickness and sense of presence were negatively related in both groups [20], the lower acceptance and appreciation of the VR experience in Estonian older adults was linked to reduced positive feelings rather than to the occurrence of symptoms [62].

Although the two studies foresaw slightly different protocols (in the Estonian study, participants had to finish the shopping task only once; instead, in Italy, participants completed as many shopping lists as possible in 20 min), given the obtained outcomes, we can hypothesize that the user experience was not influenced. If this was the case, in fact, we would have noticed shorter and possibly more engaging sessions in Estonia, whereas we noticed the opposite: the VR experience in Estonia had a median duration of 20 min for the completion of just one trial.

Estonia is a country where the use of technology and digital systems is widespread in public and private sectors [27,63], so it is plausible that participants could be more critical and demand more from technology. Moreover, being more familiar with VR could also contribute to reducing the so-called “wow” effect, which usually occurs the first time a person experiences immersive VR [64]. Therefore, Estonian social background could partly explain these results. Nonetheless, Estonian participants were possibly more cautious in the interaction with the VR environment. This can be seen in performance data: Estonian users made fewer errors than Italian ones—and the difference was not statistically significant, but needed considerably more time to complete the task. This fact may have led to lower satisfaction scores. This hypothesis would also agree with participants’ spontaneous comments, which mostly indicated that the system was difficult and uncomfortable to use, and not intuitive, or, rather, that it would have been more suitable for younger people. Nonetheless, participants appreciated the experience, found it potentially fun and engaging, and recognized its potential educational aim.

## 7. Limitations

We acknowledge that this study has some limitations. First, the sample size was small and did not allow to generalize the findings to the entire Estonian population of older adults with cognitive deficits. However, usability studies are generally conducted with smaller samples [65], and 15 participants could be considered enough to capture experience issues.

A second limitation is related to the gender distribution of the Estonian sample, in which 93% of the participants were women. Currently, the relationship between gender and subjective variables during VR experiences is debated, with some works reporting no differences and some others showing that females are more susceptible to sickness and more prone to feel present [66,67]. Nonetheless, in the previous experience, we had with the Virtual Supermarket, we found no differences depending on gender, with the only exception of spatial presence and only in the group with subjective cognitive decline.

Finally, we could not obtain an evaluation of the cognitive status of the Italian and Estonian participants with the same cognitive tests, as MCI was assessed with MMSE in Italy and MoCA in Estonia. Nonetheless, most of the research on this topic highlighted a match between scores 26–28 in the MMSE and 21–23 in MoCA [53,54], allowing us to hypothesize that the cognitive compromise was comparable.

## 8. Conclusions and Future Work

The results obtained in this work suggested that the Virtual Supermarket could be generally considered acceptable and, to some extent, enjoyable by older adults with MCI. Moreover, the comparison of the outcomes obtained between two countries with different social, cultural, and technological backgrounds showed that the user-experience may vary because of these differences, and that the relation is not always straightforward.

As an implication for practice, we thus suggest verifying that the user experience is positive each time a different population—in terms of country, illness severity, background— is enrolled in a study.

Our future work will focus on the improvement of the user experience in the Estonian participants. In particular, more attention will be paid to explaining the task and how to deal with the controller. A “tutorial” simplified environment will be created to promote familiarization prior to the real test. Moreover, hints will be added in case of hesitation. The new version will then be assessed with an acceptance study. In the case of positive outcomes, clinical studies aimed at assessing, on the one hand, the effectiveness of the Virtual Supermarket as a tool for cognitive training and, on the other, whether the performance in the shopping task is able to discriminate between healthy and MCI patients will be performed.

## Figures and Tables

**Figure 1 sensors-22-08249-f001:**
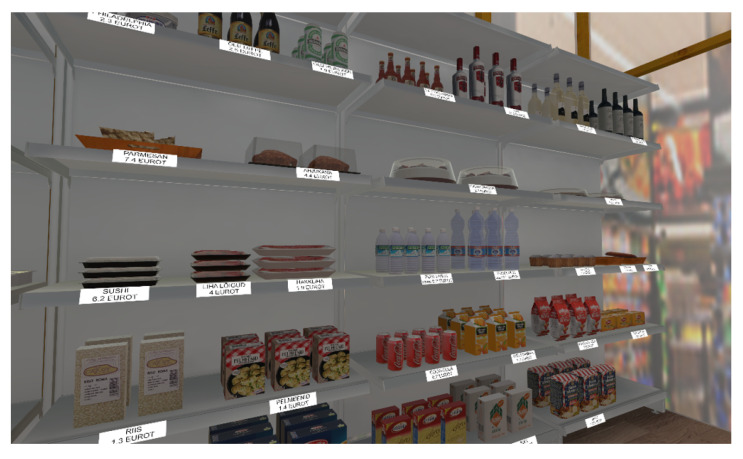
A screenshot of the virtual supermarket shelves.

**Figure 2 sensors-22-08249-f002:**
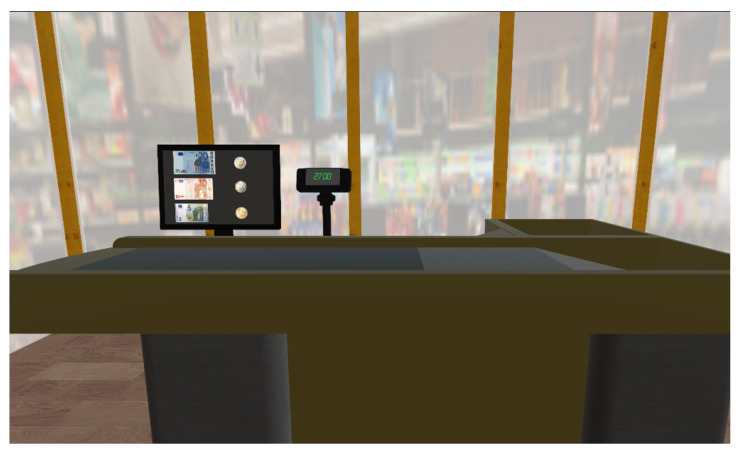
A screenshot of the virtual supermarket cash register. The amount to pay is displayed on the cash screen.

**Figure 3 sensors-22-08249-f003:**
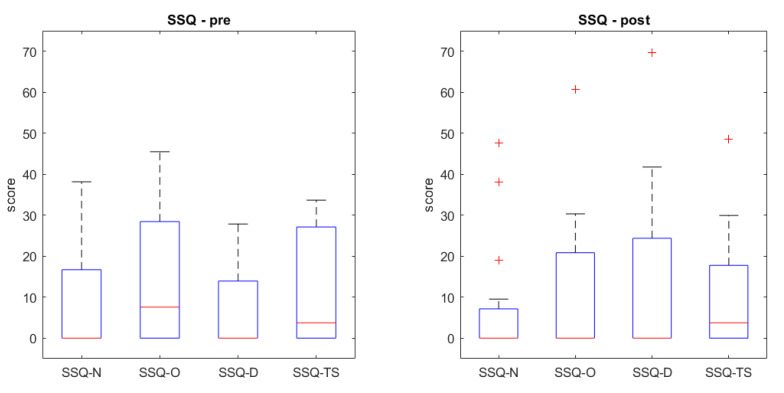
Boxplots representing SSQ distribution pre- and post- the VR experience; N: nausea, O: oculomotor disturbances; D: disorientation; TS: total score. “+” represents outliers.

**Figure 4 sensors-22-08249-f004:**
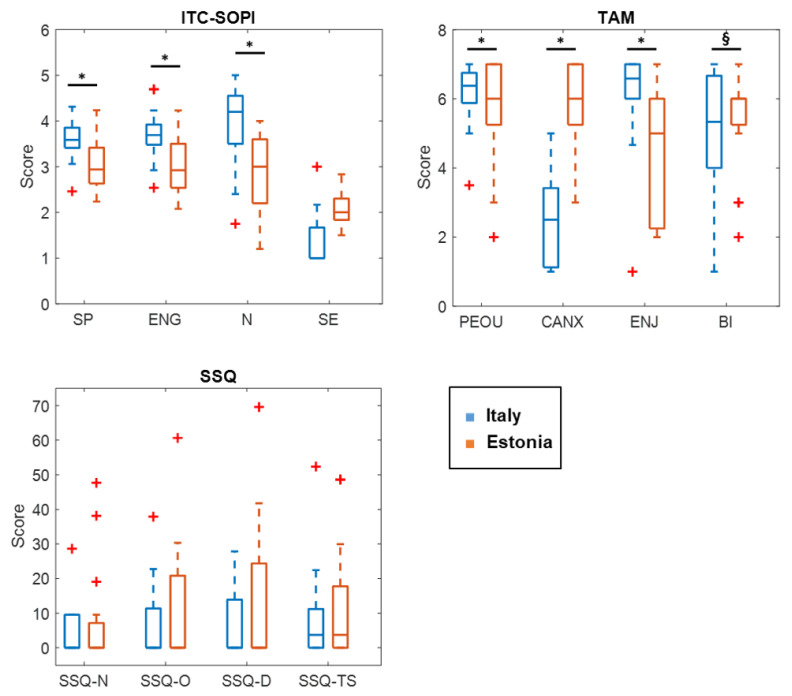
Differences in the SSQ, ITC-SOPI and TAM3 subscales between Italian and Estonian participants. * *p* < 0.05 and ${}^{§}$ *p* < 0.001. “+” represents outliers.

**Table 1 sensors-22-08249-t001:** Definition of main XR technologies (according to [11,12]) and a not exhaustive list of their applications in the context of dementia management.

Technology	Definition	Applications
Virtual reality	an artificial, virtual and viewer-centered experience that blocks out the physical world	physical activity [29], cognitive training [30], reminescence therapy [31]
Augmented reality	a hybrid experience consisting of context-specific virtual content merged into a user’s perception of the physical environment	cognitive training [32], assistance in daily life (e.g., home assistance [33] and reminders [34])
Mixed reality	an AR experience characterized by high local presence in which virtual holograms are totally blended with the real world	cognitive training [35], assistance in daily life (e.g., promotion of social engagement [36])

**Table 2 sensors-22-08249-t002:** Descriptive statistics for objective variables.

Variable	Min	Max	Mean	Std. Dev.
Time Aisle (min)	1.23	40.00	15.50	13.56
Time Payment (min)	0.5	17.46	5.09	5.70
Error: dropped item	0	3	0.93	1.10
Error: wrong item	0	1	0.21	0.43
Error: dropped and collected	0	1	0.31	0.48

## Data Availability

The data presented in this study are available upon request from the corresponding author.

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
