# Peer review of "User Experience during an Immersive Virtual Reality-Based Cognitive Task: A Comparison between Estonian and Italian Older Adults with MCI"

_sensors, 2022, doi:10.3390/s22218249_

Round 1

Reviewer 1 Report

Dear authors,

you conducted an interesting and relevant research. Some weaknesses must be fixed prior to publication:

1) Please explain dementia better for a broader, interdisciplinary readership.

2) The same with VR. Consider looking at the XR work (e.,g., http://tiny.cc/ydvzuz ) to define VR and to discuss how it differs from AR.

3) Readers might benefit from an overview table how XR technologies can be used in a medial/care setting. Please create such a table, at least with selected references and practical examples. Could be placed in section 2.

4) Not sure if it makes sense to analyze n=15 quantitatively. At least rounding the numbers would be helpful.

5) I do not see the benefits of the comparison between Estonia and Italy

6) Please reduce the number of abbreviations

7) Chapter 7 should include a more theory-driven discussion

8) Paper needs langauge polishing

9) Please highlight the research gap and contribution stronger

Internal review, not to be signed or published.

Thanks,

Reviewer

Reviewer 2 Report

The article is relevant to the mission of the journal. I consider the paper to be relevant for several reasons. It is a study that contributes to increase the field of knowledge in relation to cognitive impairment 2. It is of vital importance to provide researchers with possible lines of research in relation to the application of virtual reality. 3. It contributes to detecting factors that influence the acceptance of technology in the adult population. 4. early assessment of cognitive impairment will allow early application of cognitive training.

The topic of the paper "User experience during an immersive virtual reality-based cognitive task: a comparison between Estonian and Italian older adults with MCI" is interesting and a timely study, as it is an emerging research problem in relation to cognitive impairment and emerging technologies such as virtual reality. 

The paper is well structured, facilitating the understanding of the study. The theoretical foundation is based on the research questions, providing current and new literature in relation to the study problem and the objectives set out.

Aim: The research problem and the aim of the study are well defined.

Method: An evaluative study among adults in two countries. The acceptance and experience of adults with cognitive impairment is evaluated through the application of immersive virtual reality representing a supermarket.

The research phases are presented in a clear and structured way and the research questions are answered in a clear and detailed manner.

Results: This evaluator considers that the results shown in terms of the study problem are relevant.

The limitations are clearly specified in the study.

In short, I consider that this is a good work that will contribute to the advancement of knowledge in the field of acceptance of technology and people with cognitive impairment.

Reviewer 3 Report

Thank you for the opportunity to review your manuscript, User experience during an immersive virtual reality-based cognitive task: a comparison between Estonian and Italian older adults with MCI.

It is unnecessary to make a sub-section in the introduction to explain the above evidence.

The objective of the study is not worded.

Material and methods. It is necessary to have an order: type of study, population, sample, inclusion and exclusion criteria, etc. Now it is all very mixed up.

The registration number of the Ethics Committee is missing.

Measurement - evaluation tools need to be better explained. It is confusing to understand what and how it has been evaluated.

The statistical section is to explain how it has been analysed, not to justify (line 243)

Line 251-255- This interpretation should be discussed, not in the statistical analysis.

The beginning of the discussion should summarise the fulfilment of the study's objectives, not provide outcome data.

There is no discussion based on other studies.

The limitations section should focus on the limitations and is not a discussion.

I do not see conclusions but a summary of the objectives mixed with possible future studies.

Reviewer 4 Report

Dear authors, it is a pleasure for me to have been able to review your manuscript. I would like to make a few comments in order to improve the quality of the text. 

The objectives should be explicitly stated at the end of the introductory section.

Although the sample selection criteria are specified in the section on participants, there is no explanation of how the sample is accessed, nor who is in charge of sample collection.

It would be useful to discuss the psychometric properties of the questionnaires used.

Round 2

Reviewer 3 Report

The authors have made all the suggested changes and answered all my questions.

The work has improved sufficiently for publication.

I congratulate the authors for their efforts in modifying the suggested content.